# Chronic Fatigue Syndrome in Patients with Deteriorated Iron Metabolism

**DOI:** 10.3390/diagnostics12092057

**Published:** 2022-08-25

**Authors:** Michał Świątczak, Krzysztof Młodziński, Katarzyna Sikorska, Alicja Raczak, Paweł Lipiński, Ludmiła Daniłowicz-Szymanowicz

**Affiliations:** 1II Department of Cardiology and Electrotherapy, Medical University of Gdańsk, Dębinki 7, 80-211 Gdańsk, Poland; 2Department of Tropical Medicine and Epidemiology, Medical University of Gdańsk, Dębinki 7, 80-211 Gdańsk, Poland; 3Clinical Psychology Department, Faculty of Health Sciences, Medical University of Gdańsk, 80-210 Gdańsk, Poland; 4Department of Molecular Biology, Institute of Genetics and Animal Breeding, Polish Academy of Sciences, Wólka Kosowska, 05-552 Jastrzębiec, Poland

**Keywords:** chronic fatigue syndrome, fatigue, iron metabolism, iron overload, iron deficiency

## Abstract

Fatigue is a common, non-specific symptom that often impairs patients’ quality of life. Even though fatigue may be the first symptom of many serious diseases, it is often underestimated due to its non-specific nature. Iron metabolism disorders are a prominent example of conditions where fatigue is a leading symptom. Whether it is an iron deficiency or overload, tiredness is one of the most common features. Despite significant progress in diagnosing and treating iron pathologies, the approach to chronic fatigue syndrome in such patients is not precisely determined. Our study aims to present the current state of knowledge on fatigue in patients with deteriorated iron metabolism.

## 1. Introduction

The phenomenon of fatigue is defined as a subjective feeling of exhaustion and lack of energy, resulting in difficulties in performing daily duties [1]. It is postulated that the feeling of fatigue is the body’s defense mechanism against the excessive use of functional reserve [2], which includes physical and mental components [3]. It is estimated that 25.6% of men and 42.9% of women aged ≤36 and 33.9% of men and 66.3% of women aged >36 suffer from fatigue [4].

Fatigue is a common problem encountered in everyday clinical practice and may be one of the first symptoms of many diseases. Contrary to acute fatigue [5], which is temporary, self-limiting, and most often caused by an easily identifiable factor related to physical or mental exertion [6], chronic fatigue presents over six months, is a feature of long-lasting pathology, not strictly related to exercise, and is not relieved after rest [7]. Therefore, chronic fatigue significantly reduces patients’ quality of life [8]. Patients suffering from many diseases, such as depressive disorders, cancers, sarcoidosis, multiple sclerosis, post-stroke, heart failure, report severe chronic fatigue [9,10,11,12,13,14]. Iron deterioration (both iron insufficiency and overload) is another significant reason for chronic fatigue. In the present review, we aimed to systematize the current state of knowledge concerning fatigue in the most common diseases associated with disturbances of iron metabolism in the human body.

## 2. Methods

In preparing the content of our manuscript, we followed the Preferred Reporting Items Requirements for Systematic Reviews and Meta-Analysis (PRISMA) guidelines [15]. Relevant research studies were sourced using PubMed, Scopus, and Wiley electronic databases. A search was conducted with free-text terms for fatigue in iron deficiency diseases and fatigue in iron overload diseases. We checked source materials from 1989 to 2022 and found 296 publications. Eventually, 122 papers were selected for use in the article creation process, based on the impact of the latter studies on current patient management.

## 3. Assessment of Fatigue

Due to the common occurrence, fatigue needs to be objectified, which could allow physicians to obtain additional information about the clinical condition of a patient. It seems to be particularly useful in monitoring patients’ quality of life during disease and treatment. Many scales have successfully been developed to assess fatigue among patients in everyday clinical practice. We describe the most popular below, each characterized by a strong internal consistency, which tells us how well a test or survey is measuring analyzed parameters. Additionally, Table 1 selected scales for assessing fatigue and examples of their use in medicine.

The Fatigue Severity Scale (FSS) is one of the most popular scales used for the assessment of the severity of fatigue in many diseases. That scale evaluates the effect of fatigue on daily living. This simple tool consists of nine items, each of which is answered on a seven-point scale, where “one” means “strongly disagree”, and “seven”—“strongly agree”. The maximum number of points to be obtained on this scale is 63, and a higher score indicates greater fatigue severity. FSS is characterized by a strong internal consistency coefficient of 0.93 [16]. The FSS was successfully used in the assessment of fatigue among patients with multiple sclerosis and systemic lupus erythematosus (SLE) [13,17].

The next most frequently used scale is the Multidimensional Fatigue Inventory (MFI), consisting of twenty items. The questions included in this scale allow the assessment of five components of this symptom: General Fatigue, Physical Fatigue, Mental Fatigue, Reduced Motivation, and Reduced Activity [18]. Higher scores indicate a higher level of fatigue. MFI has good internal consistency, with an average Cronbach’s alpha coefficient of 0.84 [19]. The MFI was successfully used in the assessment of fatigue in patients with Parkinson’s disease, SLE, and many other conditions [19,20].

The next tool used to assess fatigue severity is the Fatigue Assessment Scale (FAS), which consists of ten items to determine both physical and mental fatigue. Based on the questions, the points from one to five could be obtained, where “one” corresponds to “never” and “five” to “always”. The maximum number of points to be scored on this scale is 50; if the score is between 10–20, it corresponds to a standard score and no fatigue, while 22–50 is equal to substantial fatigue. The FAS has a strong internal consistency coefficient of 0.91 [21]. The FAS was successfully used in the assessment of fatigue among patients with sarcoidosis, neurology, cardiology, as well as oncology [11,21,22,23,24].

Another scale is the Brief Fatigue Inventory (BFI), initially designed to assess fatigue among cancer patients rapidly. It consists of nine items: the first three ask patients to rate the severity of their fatigue “right now,” at its “usual” level, and its “worst” status during the past 24 h. Another six items assess how fatigue has affected daily aspects of their lives over the past 24 h [25]. The patient can receive between 0 and 10 points for each answer. The level of fatigue assessed by the BFI can be divided into categories of “mild” (1–4), “moderate” (5–6), and “severe” (7–10) pain based on the amount of pain-related interference with function. The BFI has a strong internal consistency coefficient of 0.96 [26]. The BFI is mainly used to assess fatigue among cancer patients [27].

The Functional Assessment of Chronic Illness Therapy—Fatigue (FACIT-f) is a 13-item questionnaire constructed to assess fatigue and its impact on daily activities. The questionnaire assesses physical tiredness, functional fatigue, emotional fatigue, and the social consequences of fatigue. Participants’ responses range from 0 (not at all) to 4 (very much), except for items #7 and #8, which are reversed scored. The total score range is 0–52. A score of less than 30 indicated severe fatigue, and the higher the score, the better quality of life. The FACIT-f has high internal consistency, with a Cronbach’s α coefficient exceeding 0.96 [28]. The FACIT-f is mainly used to assess fatigue among patients with cancer, lupus erythematosus, and rheumatoid arthritis [28,29,30].

The Chalder Fatigue Scale (CFQ) is an 11-item scale that provides an assessment of physical fatigue (measured by items 1–7) and mental fatigue (measured by items 8–11). Each question can be answered as follows: less than usual, no more than usual, more than usual, much more than usual. The maximum number of points to be scored on this scale is 33 (Linkert scoring method), and a higher score indicates greater severity. The reliability of this tool is high (0.9) [31]. The CFQ was successfully used in the assessment of fatigue among patients with chronic fatigue syndrome, SLE, and Primary Sjogren’s Syndrome [32,33,34].

The revised Piper Fatigue Scale (PFS) consists of 22 items that assess present fatigue [35]. The PFS measures four dimensions of subjective fatigue: behavioral/severity, affective meaning, sensory, and cognitive/mood. The patient can obtain a score for the subject from 0–10 (minimal–maximal fatigue). A total fatigue score can be calculated by adding the four subscale scores and dividing this sum by four. A strong internal consistency characterizes the PFS. The PFS is usually used to assess fatigue among cancer patients [36].

The Inflammatory Bowel Disease Fatigue Scale (IBD-F) consists of three sections. Section one measures the severity and frequency of fatigue, section two measures the experience and impact of fatigue on the individual’s life, and section three contains unscored open-ended questions about possible factors contributing to fatigue [37]. The first two sections are scored on a Likert scale from 0 to 4, and possible total scores are 0–20 in section one and 0–120 in section two. Higher scores indicate more significant fatigue and a greater impact of fatigue on quality of life.

The Multidimensional Assessment of Fatigue (MAF) scale was developed to assess fatigue among rheumatoid arthritis patients. It contains 16 questions and measures fatigue in four aspects: severity, distress, degree of interference with activities of daily living, and time [38]. The total score ranges from 1 (no fatigue) to 50 (severe fatigue), with higher scores indicating higher levels of fatigue and its impact on the person.

The Short-Form 36 Questionnaire (SF-36) provides an assessment of the patient’s quality of life based on 36 questions relating to 8 indicators of quality of life (physical functioning, limitations in fulfilling roles due to physical health, complaints of pain, general sense of health, vitality, social functioning, role limitation due to emotional problems, sense of mental health) [39]. Each indicator is transformed into a 0–100 scale. A score of 0 is equivalent to a maximum disability, and a score of 100 is equivalent to no disability.

## 4. Clinical Presentation of Iron Disturbances

In the human body, iron ions are distributed in three forms. About 60% of iron is incorporated into hemoglobin in the form of heme, which is essential for binding oxygen [40]. Another 30% is stored in the form of ferritin and hemosiderin [40]. The remaining 10% of iron ions are part of cytochromes, iron-containing enzymes, or myoglobin [40].

### 4.1. Iron Deficiency

Iron is a micronutrient necessary for the proper functioning of the body. It is responsible for functions such as binding and transporting oxygen to tissues, regulating its availability in the body using ferritin and hemosiderin, and taking part in regulating enzymes involved in cellular respiration [41]. It is also a component of myoglobin and mitochondrial enzymes; therefore, its deficiency may result in fatigue and weakness [42]. Fatigue is the main symptom of iron deficiency, described by patients as deterioration of motivation to perform daily tasks, feelings of physical tiredness, or problems with concentration. Fatigue could be presented in iron deficiency patients before they become anemic.

Iron deficiency is one of the most common nutritional deficiencies in the world [43]. It is estimated that about 1.6 billion people worldwide suffer from that pathology [44]. Among mature Europeans, its frequency ranges from 4% to 33%, while in the elderly, it ranges from 11% to over 50% [44,45]. The underlying cause of this condition can be divided into three major categories: insufficient iron supply, impaired absorption, and increased iron requirements [46]. The diagnosis of iron deficiency is based on the values of ferritin level, iron transferrin saturation (TfS), total iron-binding capacity (TIBC), and soluble transferrin receptor (sTfR) concentration. After an initial estimate of iron deficiency, an in-depth history should be taken to determine its potential cause. These causes can be subdivided by origin into gastroenterological, gynecological, nephrological, hematological, pulmonological, and those associated with coexisting chronic disease. Management is based first on eliminating the source of potential iron loss, followed by intravenous or oral iron supplementation. Table 2 presents brief information regarding the quick iron-deficiency diagnostic tools [47,48,49]. In addition, Table 3 illustrates the accuracy and reliability of iron deficiency diagnostic methods [50,51,52,53].

#### 4.1.1. The Pathophysiology of Fatigue in Iron Deficiency

The exact pathomechanism of fatigue in iron deficiency cannot be clearly defined. Among the experimental studies, several data have addressed the pathophysiology of fatigue in this condition [54,55,56]. For instance, Rineau et al. demonstrated, in mouse models, that iron deficiency without anemia causes a decrease in the activity of complex I of the respiratory chain in a mitochondrion [56]. Iron in cytochromes is part of the prosthetic group enabling the proper functioning of the complexes that make up the respiratory chain in the mitochondrion. Decreased activity of complex I of the respiratory chain resulted in a decrease in the performance of muscles composed mainly of type I and IIA fibers, which derive their energy primarily from aerobic respiration [55]. However, such an outcome was not observed when examining the performance of muscles composed mainly of type IIB fibers [56]. This may suggest that decreased iron level leads to the deterioration of muscle performance, which draws energy from aerobic processes, thus increasing fatigue. Other authors, in their mouse models, have shown that iron deficiency mainly affects maximal tissue oxygen consumption [54]. The decreased ability of tissues, especially muscle tissue, to absorb oxygen in iron deficiency can lead to an increased cardiovascular load and thus to a feeling of fatigue.

Some authors tried to assess the fatigue mechanism among humans [57,58]. Brownlie et al. [57] and Brutasaert et al. [58] analyzed the relationship of iron deficiency without anemia to fatigue levels among women and demonstrated that iron supplementation improved maximal oxygen consumption (VO_2_ max) and muscle performance, thus leading to a decrease in perceived fatigue.

#### 4.1.2. Gender Differences in Iron Deficiency Anemia

Iron deficiency anemia affects both men and women; however, there are significant gender differences in the frequency of IDA development. These differences have a multifactorial basis and can be caused by comorbidities, age, or physiological changes occurring during pregnancy. Levi et al. conducted a study among men and women from four European countries—Belgium, Germany, Italy, and Spain—in which they estimated the gender difference in the frequency of IDA [59]. The first parameter described in this study is the age at which the risk of developing IDA is highest. Among women, the highest risk was at ages 30–34, with a downward trend in the risk of developing IDA in ages 85–89. Among men, the risk increases at ages 65–69 and displays its highest values at ages greater than or equal to 95. Another parameter described is coexisting conditions or diseases. Significantly increased risk of developing IDA included patients with heavy menstrual bleeding, obesity, and pregnancy or breastfeeding. The likelihood also increases among women who do not take iron supplements during pregnancy. The risk increases along with the trimester of pregnancy. A risk factor for the development of IDA among men is mainly obesity, which is associated with poor diet and reduced iron supply with food. Moreover, obese patients are characterized by persistent inflammation, which plays a key role in the development of IDA. The presence of comorbidities such as gastritis, peptic ulcer disease, esophagitis, gastric cancer, and inflammatory bowel disease increase the risk of developing IDA in both sexes. However, there has been a trend toward a higher incidence of IDA in women with artificial heart valves, most likely due to the development of postoperative complications such as perivalvular leakage and subsequent hemolysis, which develops due to turbulent retrograde blood flow across the valve. Among the accompanying symptoms of IDA, headaches, a feeling of weakness, and hair loss were observed to predominate in women, while a feeling of weakness and headaches predominated in men.

#### 4.1.3. Iron Deficiency and Fatigue in Women of Reproductive Age

Iron deficiency is a common presentation in women of reproductive age; therefore, fatigue could be a frequent symptom for that population (Table 4). Some data in the literature address that problem, revealing the relief of fatigue after iron supplementation. A meta-analysis by Greig et al. summarized several papers in which the studied population’s measured fatigue was initially higher in the iron-deficient subgroup and decreased during iron supplementation [60]. Similar results were obtained in another study that used the earlier described PFS scale to assess fatigue levels [61]. Evaluation of the relationship between cognitive function and iron deficiency also showed improvement after iron supplementation; however, it cannot be determined which components of cognitive function are enhanced [60]. Holm et al. compared the effects of intravenous versus orally administered iron therapy among women who experienced postpartum hemorrhage and observed a significant reduction in physical fatigue levels in both groups and a more rapid improvement for the group that received intravenous iron [62].

#### 4.1.4. The Importance of Fatigue among Patients with Heavy Menstrual Bleeding (HMB)

Women with HMB tend to lose significantly more iron per menstrual cycle (5–6 times more) compared to normal menstruating women, which can lead to faster depletion of iron stores (Table 4) [63]. Excessive blood loss because of HMB can lead to complications such as iron deficiency with or without accompanying anemia or hemodynamic instability, and fatigue is the most common.

There are limited sources in the available literature that relate to the feeling of fatigue among female patients with HMB. Wang et al., in their study performed on a small group of women with diagnosed HMB between 10 and 17 years of age, showed that fatigue estimated by the described earlier FSS scale was significantly higher than in the female control group [64].

Kocaoz et al. evaluated iron impairment among women with HMB based on hemoglobin and ferritin levels, and fatigue levels were assessed using the mentioned-above SF-36 QoLS scale (Quality of Life Scale) [65]. This study revealed that lower values of these two laboratory parameters were observed amongst women suffering from HMB, which was associated with increased feelings of fatigue and difficulty in performing daily activities such as general activity, mood, ability to walk, ability to perform basic household tasks (such as washing, laundry, cleaning), relationships with other people, and enjoyment of life [65]. There was also a weak negative correlation between menstrual duration and daily activities and a weak positive correlation between menstrual time and perceived health on the SF-36 QoLS scale [65].

#### 4.1.5. Fatigue in Anemia of Chronic Diseases

Many chronic diseases could be associated with anemia and fatigue (Table 4). Chronic kidney disease (CKD), inflammatory bowel disease (IBD), and chronic heart failure (CHF) are prominent examples. Table 2 presents the changes in laboratory parameters observed in anemia of chronic disease.

Fatigue is a typical feature in CKD patients. The cause of fatigue in CKD is multifactorial and is based on decreased oxygen delivery, which leads to increased anaerobic metabolism in tissues, increased lactic acid production, and eventually lactic acidosis. Metabolic acidosis and accompanying hyperphosphatemia lead to decreased muscle strength and thus feelings of physical fatigue. Increased urinary protein loss accompanying chronic kidney disease can lead to sarcopenia and to feelings of physical fatigue. It is also important to remember that chronic diseases are very often accompanied by depression, which can contribute to feelings of mental fatigue [66]. In the study, named FERWON (stands for Trial of Intravenous Iron Isomaltoside/Ferric Derisomaltos and Iron Sucrose), which included 3050 patients with CKD and IDA, fatigue was assessed using the Functional Assessment of Chronic Illness Therapy Fatigue Scale (FACIT—F) in two subgroups: FERWON-NEPHRO and FERWON-IDA patients [67,68]. The first group—FERWON-NEPHRO (Iron Isomaltoside/Ferric Derisomaltose vs. Iron Sucrose for Treatment of Iron Deficiency Anemia in Non-Dialysis-Dependent Chronic Kidney Disease), included adult subjects with chronic kidney disease and known iron deficiency anemia, and the second group—FERWON-IDA (Iron Isomaltoside/Ferric Derisomaltose vs. Iron Sucrose for the Treatment of Iron Deficiency Anemia), included adult patients who had IDA with associated intolerance to oral iron therapy. This study was designed to evaluate the safety and efficacy of different medications in iron supplementation treatment in two substudies, showing a significant decrease in fatigue levels within the treatment. In the beginning, more than half of the patients presented high levels of fatigue at baseline. In the FERWON-IDA study, either treatment with iron isomaltoside/derisomaltoside or iron sucrose reduced fatigue levels by about 15 points on the FACIT scale from baseline. Moreover, the improvement was faster with iron isomaltoside/derisomaltoside treatment. In the FERWON-NEPHRO study, iron isomaltoside/derisomaltoside treatment and iron sucrose treatment led to improvements of >10 points on the FACIT fatigue scale at week 8 [68].

CKD also leads to erythropoietin (EPO) deficiency. EPO deficiency in patients suffering from chronic kidney disease is due not only to the disease but also to the accompanying chronic inflammation. Moreover, Mercadal et al. observed that anemia in CKD patients with GFR >30 mL/min is most often caused by factors other than EPO deficiency [69]. However, as the disease progresses, the decline in EPO levels increases, leading to an exacerbation of the existing anemia.

IBD is the next typical example of chronic disease with anemia. Fatigue in IBD is one of the predominant symptoms, along with abdominal pain and diarrhea [70]. Currently, three scales can be used to measure fatigue levels among IBD patients: Inflammatory Bowel Disease Fatigue Scale (IBD-F), Multidimensional Fatigue Inventory (MFI) scale initially used to measure fatigue among patients undergoing chemotherapy and individuals who have chronic fatigue syndrome, and the third scale, the Multidimensional Assessment of Fatigue (MAF) scale, which was developed to assess fatigue among rheumatoid arthritis patients. Each scale is described in an earlier section of the article. Norton et al., in their study, demonstrated that both the MAF and MFI scales could be successfully used to assess fatigue in patients with IBD [38]. It is estimated that 21–41% of patients with IBD are in remission, and 70–75% of patients with active IBD experience fatigue [71,72].

The available literature does not provide clear answers regarding the causes of fatigue in IBD or the relationship between fatigue severity and iron deficiency. For instance, Goldenberg et al. showed no relationship between fatigue and iron deficiency in patients without anemia [70]. In contrast, Jonefjäll et al. narrowed their study to only subjects with ulcerative colitis and revealed that low ferritin levels are related to higher fatigue intensity [3].

CHF is another critical example of chronic disease anemia. The pathomechanism of fatigue in patients with CHF is multifactorial and is based on impaired absorption of macro- and microelements, including iron due to gastrointestinal edema, increased gastrointestinal blood loss due to anticoagulants, and inflammation that prevents the use of stored iron for erythrocyte production [73]. The FAIR-HF study (which stands for Ferinject Assessment in Patients with Iron Deficiency and Chronic Heart Failure), was conducted on 459 adult patients with NYHA class II-III, ferritin levels less than 100 ug/L or less than 300 ug/L with associated TSAT < 20% and hemoglobin levels between 9.5 and 13.5 g/dl, aimed to evaluate the efficacy of iron supplementation in improving CHF symptoms, including fatigue in patients with iron deficiency [74]. The study showed that intravenous iron supply was associated with significant improvement in both Patient Global Assessment (which is the self-assessment of scale) and NYHA classification scores [74]. In addition, 6-min walk test scores and quality of life improved significantly [74]. The CONFIRM-HF study (which stands for Ferric CarboxymaltOse evaluatioN on perFormance in patients with IRon deficiency in coMbination with chronic Heart Failure) observed similar results regarding fatigue levels were observed [75].

#### 4.1.6. Fatigue in Aplastic Anemia, Nocturnal Paroxysmal Hemoglobinuria, and Myelodysplastic Syndrome

Aplastic anemia, paroxysmal nocturnal hemoglobinuria, and myelodysplastic syndrome (MDS) are rare diseases associated with impaired bone marrow function in producing abnormal morphotic elements or reduced or inhibited the production of these elements (Table 4). The symptoms of these diseases are varied, but a widespread sign is a feeling of fatigue. The pathophysiology of fatigue in these diseases is multifactorial and may be related to medications, emotional stress, sleep disturbances, or anemia [76].

Oliva et al. observed a significant reduction in quality of life among patients with diagnosed aplastic anemia, paroxysmal nocturnal hemoglobinuria, and MDS, requiring intravenous iron supply, which correlated with hemoglobin levels and several comorbidities [77]. In addition, MDS patients reporting high fatigue levels were characterized by significant disease symptom severity and lower quality of life [8]. It has been observed that fatigue levels reported by MDS patients provide an even more accurate assessment of the quality of life than hemoglobin levels [8].

#### 4.1.7. Fatigue in Elderly Population

The pathophysiology of fatigue in the elderly is multifactorial and different in various patients (Table 4). During the aging process, the immune system impairs, thus leading to low-grade chronic inflammation, impairment of multiple endocrine regulatory loops, and increased susceptibility to developing chronic diseases [78]. The accompanying dysregulation of body homeostasis in the elderly leads to impaired gastrointestinal iron absorption and decreased iron availability due to inflammation and increased hepcidin levels [79]. Finally, the lack of iron supplementation among these individuals could exaggerate the iron deficiency in the body [80]. Neidlein et al. conducted a study among 224 individuals aged 65–95 years who were hospitalized for various reasons [42]. In this study, almost 50% of the hospitalized patients had concomitant iron deficiency, with most cases (86%) presenting functional iron deficiency, defined as adequate iron stores but insufficient availability of iron for incorporation into erythrocyte precursors [42]. Moreover, concomitant anemia was predominant among the group with functional iron deficiency (72%) [42]. When comparing the groups of patients with and without iron deficiency, significantly higher fatigue levels were observed in patients with iron deficiency, lower Hb levels, and higher CRP levels [42]. It was observed that patients diagnosed with iron deficiency who received iron supplementation during hospitalization were characterized by improvements in isometric exercise and generalized improvements in performance status [70,81,82,83]. In addition, Houston et al. demonstrated a significant impact of iron supplementation on reducing experienced fatigue in iron deficiency in elderly patients [84].

**Table 4 diagnostics-12-02057-t004:** The iron deficiency clinical conditions discussed in the article with fatigue as one of the leading symptoms.

Name of Clinical State	Brief Description of Presented Clinical States
Iron deficiency and fatigue in women of reproductive age	Women of childbearing age are particularly vulnerable to iron deficiency due to increased requirements during pregnancy and the loss of this nutrient during menstruation or childbirth. Other causes of iron deficiency may be an inadequate diet low in iron and high in substances that inhibit iron absorption from the gastrointestinal tract. Iron deficiency among women of reproductive age is a common phenomenon observed worldwide. In the USA, Japan, and Europe, the prevalence is 10–20% [85,86].
Fatigue among patients with heavy menstrual bleeding	According to the NICE (National Institute for Health and Care Excellence’s) definition, it is excessive blood loss at the time of expected menstruation that disrupts the physical, emotional, social, and material elements of a woman’s quality of life, which may occur alone or in combination with other symptoms [65]. The prevalence of this disorder is estimated to be between 27.2% and 54% amongst young women [87,88,89].
Anemia of chronic diseases	Anemia can have various origins; however, it is mostly caused by iron deficiency or chronic disease. Iron deficiency anemia is characterized by decreased hemoglobin synthesis, leading to the development of microcytic and hypochromic erythrocytes [90]. In contrast, anemia of chronic diseases is characterized by the normal iron content in the body with inadequate iron distribution, leading to the development of normocytic and normochromic erythrocytes [91]. Moreover, in the anemia of chronic diseases, the current inflammation results in decreased production and release of EPO, which is responsible for enhancing erythrocyte formation in the bone marrow [92]. Inflammatory bowel disease: includes Crohn’s disease and ulcerative colitis. The etiology of these diseases has not been clearly explained, and it is currently assumed that in genetically predisposed individuals, certain environmental factors trigger a generalized immune response leading to the development of symptoms both from within and outside the gastrointestinal tract. The most common symptoms are abdominal pain, diarrhea, fatigue, perianal lesions, osteoarticular, and skin symptoms. IBD diagnosis is based on recognizing clinical signs and characteristic changes in imaging studies. The treatment of IBD is mainly based on mesalazine or sulfasalazine and glucocorticosteroids to induce remission, followed by mesalazine or thiopurines to maintain remission. Chronic heart failure: a pathological entity in which cardiac function is insufficient to ensure adequate delivery of oxygen and nutrients to tissues or an increased filling pressure occurs to maintain sufficient minute capacity. It is estimated that between 37% and 61% of patients with chronic heart failure have concomitant iron deficiency, with or without anemia [93]. It contributes to the development of symptoms such as fatigue, decreased exercise tolerance, and reduced quality of life.
Hematological disorders	Myelodysplastic syndrome: it is a hematopoietic malignancy in which ineffective hematopoiesis with features of dysplasia of ≥1 hematopoietic cell line and ≥1 cytopenia in the peripheral blood is observed. The incidence of this disease is 4 cases per 100,000 [94]. The most common symptom is anemia, which is often associated with fatigue, weakness, exercise intolerance, chest pain, or cognitive impairment [95]. Aplastic anemia: it is a rare disease characterized by marrow hypoplasia or aplasia, often secondary to pluripotent stem cell injury. The causes of the development of aplastic anemia can be divided into rare congenital causes, such as Fanconi anemia, or more common acquired causes, such as exposure to ionizing radiation. The predominant symptoms in such patients are fatigue, pallor, headache, dyspnea, palpitations, bleeding gums, and petechiae. Paroxysmal nocturnal hemoglobinuria: it is a rare acquired hematopoietic stem cell disorder associated with a defect in the erythrocyte cell membrane that prevents the cell from protecting itself from cytolysis. The incidence ranges from 1 to 10 cases per million but is most likely underestimated because some patients may remain undiagnosed [96,97]. Fatigue, dyspnea, and hemoglobinuria are the most common presenting symptoms.
Elderly	WHO (World Health Organization) defines multimorbidity as the presence of two or more chronic diseases. It is estimated that this condition may affect up to 95% of people aged ≥65 years [98]. The risk factors for multimorbidity are unknown, but it is speculated that aging may be one of the significant factors. Frailty syndrome is another significant risk factor. It is a syndrome occurring in chronically ill patients that includes unintentional weight loss (≥5 kg per year), fatigue, muscle weakness, slowed gait, and low physical activity. Frailty syndrome is associated with chronic inflammation of unknown cause leading to fatigue, decreased muscle mass, and decreased activity.

### 4.2. Iron Overload Disorders

Excess iron levels in the serum may be associated with fatigue; however, it has different pathophysiology than iron deficiency [99,100].

Many different factors can lead to an overload of iron in the body. Increased absorption of dietary iron, frequent transfusions, and ineffective erythropoiesis [101,102], and the most often reason is hereditary hemochromatosis (HH). HH is a genetic disease that increases iron accumulation in body tissues [100]. Besides HFE-related hemochromatosis, other iron overload mutations are not related to the HFE gene [100].

The patient should undergo a laboratory diagnosis if iron overload is suspected based on symptoms such as chronic fatigue, dark skin color, arthralgia, or hepatomegaly. If their ferritin level is >200 ng/mL for premenopausal women, >300 ng/mL for men and post-menopausal women, and transferrin saturation level is >45%, the patient should have genetic diagnostics performed to exclude the presence of mutations within the HFE gene (Figure 1) [100].

#### 4.2.1. The Pathophysiology of Fatigue in Iron Overload

The toxic effect of iron seems to be significant. It generates oxidative stress, damaging many organs and promoting the pathogenesis of neurodegenerative diseases [100,103]. In addition, too high iron levels in the human body may lead to a gradual disruption of the functions of endocrine glands, leading to diabetes, hypothyroidism, and hypogonadism, which may also result in the feeling of fatigue [104]. In the course of iron overload, cardiac dysfunction and the development of heart failure also occur [100,105].

There is a lack of reports in the literature about fatigue in iron overload diseases, and the pathophysiology of this symptom during iron overload has not been clearly defined.

#### 4.2.2. The Clinical Implications of Iron Overload

To evaluate the clinical status and the potential causes of chronic fatigue more accurately in patients with iron overload, the diagnosis should be expanded. One of the organs occupied in the course of iron accumulation is the liver. Therefore, it is crucial to assess its potential damage by determining serum aminotransferase and lipid levels and to exclude iron accumulation in the liver by, for example, biopsy [106,107,108]. Another organ that can be damaged in the course of iron overload diseases is the heart. To assess left ventricular dysfunction, echocardiography with acoustic marker tracking (STE) should be performed; additionally, cardiac magnetic resonance imaging is an accurate method for assessing heart muscle function [105,109]. As a result of pancreatic damage, patients may present with abnormal glucose tolerance or diabetes, with elevated glucose levels found in laboratory tests [110].

#### 4.2.3. Fatigue in Hereditary Hemochromatosis

Nowadays, one of the early symptoms noticed by patients with HH is the feeling of severe, chronic fatigue, which very often significantly decreases the quality of life (Table 5). Thus far, there are no data in the literature regarding the objectification of the level of fatigue in these patients, as well as the use of fatigue assessment in monitoring the effectiveness and efficiency of treatment of patients with HH.

In one of our previous papers, we presented the case of a 42-year-old man with a genetically confirmed diagnosis of hereditary hemochromatosis, in whom we observed a significant reduction in fatigue in each of the scales used in the follow-up (in FAS from 25 to 17 points, in CFQ from 12 to 5, in FSS from 37 to 13) during 6 months of treatment with venesections [109].

#### 4.2.4. Fatigue Related to Frequent Transfusions and Ineffective Erythropoiesis

There are a few erythroid disorders with ineffective erythropoiesis, which may cause an iron overload due to frequent transfusions: thalassemia syndromes (thalassemia major and intermedia), sideroblastic anemias, congenital dyserythropoietic anemias (e.g., hereditary spherocytosis and sickle cell anemia) (Table 5) [101]. It should also be emphasized that this situation can also occur in myelodysplastic syndrome and aplastic anemia, which have been described in connection with iron deficiency [104,111].

The symptoms of mentioned diseases, particularly fatigue, result from anemia, which develops in many patients and often requires frequent transfusions. The transfusions, in combination with ineffective erythropoiesis, result in iron accumulation in the body [101,111]. Currently, there is a lack of papers in the literature addressing fatigue levels concerning diseases with iron overload. Differentiating the source of this symptom is particularly difficult in conditions where the primary problem is anemia and iron overload is secondary to medical treatment such as transfusions.

While et al. show that the leading symptom of sickle cell anemia is chronic fatigue and complaints of pain [112]. In the context of fatigue therapy, Tabei et al. demonstrated that patients with beta-thalassemia minor anemia have low levels of free and total plasma carnitine and reported the efficacy of carnitine and folic acid supplementation in reducing fatigue and muscle weakness in these patients [113]. They studied 73 children with beta-thalassemia minor, evaluating their laboratory parameters and the level of plasma-free carnitine, total carnitine, red blood cell folate, and hemoglobin. Fatigue was assessed by the number of stairs climbed by the subject in the study period in a day.

To reduce the probability of developing iron overload, the frequency of transfusions must be selected very carefully, not more often than required by the actual condition of the patient and the symptoms presented by them [104]. The use of iron chelators should be considered when reducing the frequency of transfusions is not effective.

**Table 5 diagnostics-12-02057-t005:** The iron overload clinical conditions discussed in the article, in which one of the leading symptoms is fatigue.

Name of Disease	Brief Description of the Disease
Hereditary hemochromatosis	It is a genetic disease in 80% based on HFE-gene mutation, leading to increased accumulation of iron in body tissues, resulting in the generation of oxidative stress and damage to many organs. Cirrhosis, diabetes, and dark skin color were the main symptoms of HH before the era of HFE gen revealing [100]. An introduction of genetic tests in patients with abnormal iron management parameters to routine clinical practice makes it possible to diagnose HH early before the patients demonstrate symptoms of advanced diseases. Instead of the above-mentioned classic triad, one of the early symptoms noticed by patients with HH is the feeling of severe, chronic fatigue, which very often significantly decreases the quality of life. The treatment of choice for hereditary hemochromatosis are venesections. Treatment performed at the appropriate frequency significantly reduces fatigue, ferritin, and iron [109]. Venesections are more effective in reducing iron levels than chelating drugs [100]. Additionally, the applied treatment significantly improves the function of the heart muscle [109,114].
Beta-thalassemia	It is one of the genetically determined (mutation of genes located on chromosome 11) hemolytic anemia resulting from a disturbance in the synthesis of hemoglobin beta chains [115]. There are three groups of β-thalassemia: minor, intermedia, and major. Beta thalassemia major is the most severe form of beta-thalassemia. Severe symptoms of hemolytic anemia may appear already after six months of age and require treatment by regular red blood cell transfusions and sometimes chelation therapy [115]. Beta-thalassemia intermedia is a milder form of the disease.
Sideroblastic anemia	It is a group of congenital and acquired diseases characterized by the presence of peripheral microcytic anemia in the blood and an image of ring sideroblasts in the bone marrow (iron overloaded mitochondria surround erythroblast nuclei) [116].
Congenital dyserythropoietic anemias (CDA)	It is a group of rarely diagnosed anemia of unknown etiology, characterized by increased ineffective erythropoiesis, multinuclear erythroblast nuclei in the marrow, and secondary iron accumulation in tissues. Abnormal erythroblasts are destroyed in the bone marrow, resulting in elevated serum bilirubin and LDH levels [117]. The classification considers three types of CDA. In types I and III, CDA macrocytes are present, while type II CDA is characterized by normocytosis. Anemia is usually mild to moderate and manifests itself at different times in life.

## 5. Discussion

The feeling of fatigue in iron-deficient patients has a multifactorial origin, which depends on the presence or absence of comorbidities. Among individuals without comorbidities, the most important causes of fatigue secondary to iron deficiency are decreased hemoglobin concentration or dysfunction of the respiratory chain in the mitochondrion. The following pathomechanisms of the feeling of fatigue among these people can be distinguished. In the case of accompanying blood loss, there is a reduction in the concentration of hemoglobin, which is essential for supplying tissues with the oxygen required for the process of cellular respiration; therefore, because of the reduction in its concentration, there is an accumulation of metabolites of anaerobic cellular respiration such as lactic acid, which, leading to the development of acidosis, impairs the functioning of tissue-building cell enzymes [118]. It should also be remembered that iron is an essential element of numerous enzymes, or cytochromes, especially cytochrome I, which is part of the respiratory chain in the mitochondrion. This is an important element in the analysis of the potential pathomechanism of fatigue, as it has been observed that a decrease in the activity of cytochrome I lead to a decrease in the efficiency of muscles composed mainly of type I and IIA fibers and thus to the feeling of fatigue [55]. The presence of comorbidities introduces further causes of iron loss, among which the previously mentioned chronic inflammation, increased blood loss, reduced gastrointestinal absorption surface or increased protein loss can be distinguished. Increased protein loss in patients suffering from chronic renal failure ultimately leads to sarcopenia, which, due to loss of muscle mass and deterioration of skeletal muscle function, leads to a feeling of fatigue [66]. On the other hand, a reduction in the absorption area of the gastrointestinal tract in diseases such as Crohn’s disease or ulcerative colitis leads not only to impaired absorption of iron but also of other macro- and micronutrients necessary for the proper functioning of the body, which, among all symptoms, also leads to fatigue [119].

As in iron deficiency, fatigue occurring in the case of iron excess is also a multifactorial phenomenon. During long-term iron overload, damage to many tissues and organs occurs due to the generated oxidative stress. Fatigue is a common symptom presented by patients with chronic liver disease, where iron overload can lead to liver fibrosis and even cirrhosis. The pathophysiology of fatigue in liver disease is not clearly understood; it appears to involve changes in central neurotransmission, which result from signaling between the diseased liver and the brain, and changes in extracerebral neurotransmission affecting fatigue sensing behavior [120,121]. Free radicals also damage pancreatic tissue, which can lead to the development of diabetes [110]. Disrupted blood glucose metabolism may result in hyperglycemic episodes, hypoglycemia, or blood glucose fluctuations, which can lead to fatigue. In addition, multi-organ complications arising in the course of diabetes can have an impact on this symptom [122]. Free iron ions may damage mitochondrial and nuclear DNA leading to the activation of fibroblast proliferation and differentiation to myofibroblasts responsible for heart fibrosis, sometimes even leading to the development of heart failure, one of the leading symptoms of which is a feeling of severe fatigue. Oxidative stress also results in impaired relaxation and delayed contraction of the heart muscle by the inhibition of SERCA2 enzyme activity leading to the increase in cytoplasmic concentration of calcium ions in cardiomyocytes [100].

Iron, being a component of many important biochemical pathways in the human body, significantly affects its functioning. Clinical conditions leading to disturbances in iron metabolism can result in many complications, such as a feeling of chronic fatigue impairing patients’ quality of life. To gain more understanding of the role of iron in the pathomechanisms of the development of this symptom, more scientific studies are needed, such as an attempt to compare non-transferrin-bound iron levels with fatigue levels in patients presenting with iron overload. In addition, to reliably assess the clinical status of patients with iron metabolism disorders, it would seem important to evaluate the treatment leading to iron level equalization using fatigue assessment scales.

## 6. Summary

Chronic fatigue is one of the most common symptoms encountered in everyday clinical practice, which often significantly reduces the quality of life. The issue of fatigue in diseases associated with the deterioration of iron metabolism in the body, especially regarding iron overload, is a significant problem for modern medicine and requires more research to explore this issue. Validation of appropriate fatigue assessment scales in diseases with iron overload is needed.

## Figures and Tables

**Figure 1 diagnostics-12-02057-f001:**
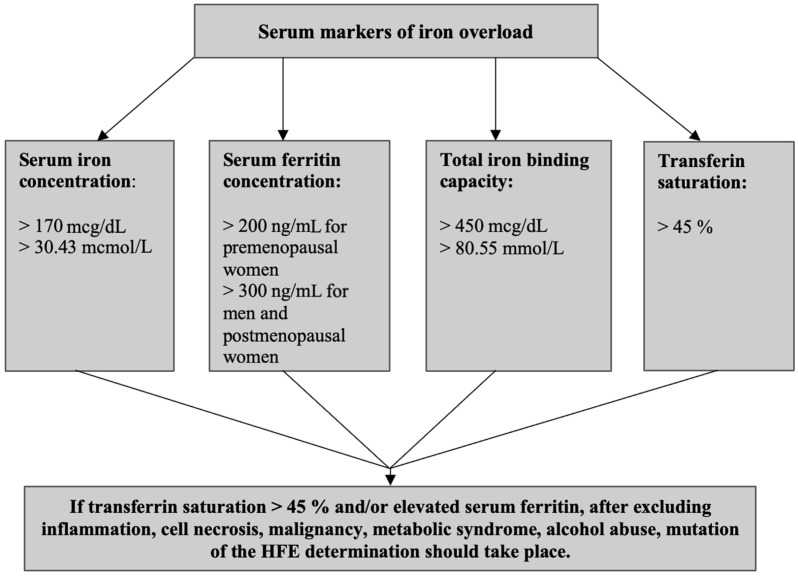
Diagnostic algorithm for iron overload diseases.

**Table 1 diagnostics-12-02057-t001:** Fatigue assessment scales and examples of their use in medicine.

Name of the Scale	Examples of Use
Fatigue Severity Scale (FSS)	Multiple Sclerosis Systemic Lupus Erythematosus Hereditary Hemochromatosis
Multidimensional Fatigue Inventory (MFI)	Parkinson’s disease Systemic Lupus Erythematosus Inflammatory Bowel Disease Cancer
Fatigue Assessment Scale (FAS)	Sarcoidosis Hereditary hemochromatosis Heart failure Stroke Cancer
Brief Fatigue Inventory (BFI)	Cancer
Functional Assessment of Chronic Illness Therapy—Fatigue (FACIT-f)	Cancer Systemic Lupus Erythematosus Rheumatoid Arthritis
Chalder Fatigue Scale (CFQ)	Chronic Fatigue Syndrome Systemic Lupus Erythematosus Primary Sjogren’s Syndrome Hereditary Hemochromatosis
Revised Piper Fatigue Scale (PFS)	Cancer Fatigue in women of reproductive age
Inflammatory Bowel Disease Fatigue Scale (IBD-F)	Inflammatory Bowel Disease
Multidimensional Assessment of Fatigue (MAF)	Inflammatory Bowel Disease
Short-Form 36 Questionnaire (SF-36)	Heavy Menstrual Bleeding

**Table 2 diagnostics-12-02057-t002:** Changes in iron metabolism parameters in iron deficiency diseases.

	Serum Iron (umol/L)	Ferritin Concentration (ug/L)	Iron Transferrin Saturation (%)	Total Iron-Binding Capacity (umol/L)	Serum Soluble Transferrin Receptor Concentration (mg/L)
Normal values	10–30	10–200 in women; 15–400 in men	17–44%	40–80 in women; 45–75 in men	1.9–4.4 in women; 2.2–5 in men
Iron deficiency anemia	↓	17–44%	↓	↑	↑
Latent iron deficiency	Normal/↓	40–80 in women; 45–75 in men	Normal/↓	↑	↑
Iron deficiency anemia refractory to iron therapy	↓	1.9–4.4 in women; 2.2–5 in men	↓	↑	↑
Anemia of chronic diseases	↓	10–30	↓	↓	Normal

↓—reduced level; ↑—elevated level.

**Table 3 diagnostics-12-02057-t003:** The accuracy and reliability of iron deficiency diagnostic methods.

Iron Metabolism Parameter	Sensitivity and Specificity
Serum iron (SF)	In the diagnosis of IDA, it provides a specificity of 38.67% and a sensitivity of 63.5% [50].
Ferritin concentration	Serum ferritin at a cut-off limit of 41 ng/mL has a sensitivity and specificity of 98% and 98%, respectively [51].
Iron transferrin saturation	There is a lack of data in the literature regarding sensitivity and specificity in IDA. In a study by Low et al. conducted in 1997, it was shown that for transferrin saturation values <20%, the sensitivity was 74%, and the specificity was 36% [52].
Total iron-binding capacity	Provides sensitivity of 64.5% with a specificity of 42.8% in the diagnosis of IDA [50].
Serum soluble transferrin receptor concentration	A study of 72 patients with advanced IDA performed by Choi et al. revealed sensitivity and specificity of 70.8% and 90.6%, respectively [53].

## Data Availability

All articles used in the process of creating this paper can be found in the PubMed, Scopus, and Wiley electronic databases.

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
