# Peer review of "Chronic Fatigue Syndrome in Patients with Deteriorated Iron Metabolism"

_diagnostics, 2022, doi:10.3390/diagnostics12092057_

Round 1
Reviewer 1 Report
This review is certainly relevant. However, there are several points, minor and major, that the authors need to consider before the manuscript can be accepted for publication:
1. A range of biomedical analytical techniques are mentioned for diagnosing iron deficiency. What are the accuracy and relability of these methods?
2. The authors should more clearly describe the Prisma system.
3. Hemoglobin is essential as an iron source. This should be included already in section 4.1. It actually covers for 60% of total iron.
4. Already in 4.1. should the gender perspective be included.
5. EPO is related to CKD (in 4.1.4). A discussion on this matter should be added.
6. Introduce the abbreviation QOL (Quality of Life)
7. Table 3. "Elderly" is not a disease.
8. Avoid the word "big". What is a "big study?"
Author Response
Dear Reviewer, as the authors of the article, we are very grateful for the reliable review and valuable comments on the content of our paper. We would also like to thank you for the valuable time you devoted to reviewing our article. We would like to refer to your specific comments: 1. “A range of biomedical analytical techniques are mentioned for diagnosing iron deficiency. What are the accuracy and relability of these methods?” We have introduced a new table into the article: "Table 3. The accuracy and reliability of iron deficiency diagnostic methods.". We hope that the information contained therein will coherently complement the content of our paper. 2. “The authors should more clearly describe the Prisma system.” We followed the Preferred Reporting Items Requirements for Systematic Reviews and Meta-Analysis (PRISMA) whenever possible. The structure of our article made it difficult to follow each point of the PRISMA guidelines. A search in electronic databases (PubMed, Scopus, and Wiley) was conducted with free-text terms for fatigue in iron deficiency diseases, and fatigue in iron overload diseases. 3. “Hemoglobin is essential as an iron source. This should be included already in section 4.1. It actually covers for 60% of total iron.” We have expanded paragraph 4.1 by touching on the iron sources in its introduction. 4. „Already in 4.1. should the gender perspective be included.” We have introduced a new paragraph into the article: “4.1.2. Gender differences in iron deficiency anemia”. 5. “Introduce the abbreviation QOL (Quality of Life)” Thank you for noticing this oversight, we have clarified the abbreviation QOL in the text of the article. 6. “Table 3. "Elderly" is not a disease.” We have renamed the column of Table 4 to 'Name of clinical state', so it no longer suggests that “Elderly” is a disease. 7. “Avoid the word "big". What is a "big study?"” We have reworded the paragraph where we used the statement 'big study'. Thank you very much for every important suggestion. All comments helped us to improve the quality of our article. We have tried to follow all the comments of Reviewers, trying to increase the value of our paper. We hope that we have met your expectations regarding the revision of our article. Sincerely yours, Authors.
Reviewer 2 Report
In the present article the authors wish to draw attention to the link between fatigue as a symptom and abnormalities of iron metabolism. This relationship is well known to clinicians and therefore during a clinical examination presenting fatigue, the search for markers of iron metabolism should be performed.
The authors successively describe how to assess a patient's fatigue and iron metabolism abnormalities (some causes of deficits and accumulation), however, no proposal is really original in the diagnosis nor in the management of the patient.
The authors conclude that the pathophysiology remains a mystery, but they do not propose any research on the subject either.
Author Response
Dear Reviewer,
The authors wish to thank for the professional and most valuable comments provided by the Reviewer. The Reviewer introduced further essential new aspects and allowed the authors to approach their data with ample criticism.
We have inserted a new paragraph into the article: "4.2.2 The clinical implications of iron overload", in which we try to introduce additional diagnostic tests among patients with iron overload, the results of which may give us an idea of the starting point of increased fatigue in patients with this type of condition. In addition, we have expanded the information on the diagnosis of iron deficiency to include its potential causes.
All comments helped us improve the quality of our article. The authors tried to answer the suggestion with due attention and diligence to the limit of their study data.
Sincerely yours,
Authors.
Round 2
Reviewer 1 Report
The manuscript can now be accepted for publication.
Author Response
Dear Reviewer,
thank you very much for the valuable time you took to re-evaluate our article. We are very happy that our corrections incorporating your comments met your expectations. Thank you for accepting our paper.
Sincerely yours,
Authors.
Reviewer 2 Report
The text has not really been improved.
There is no explanation of what is the link between iron metabolism and patient fatigue.
Author Response
Dear Reviewer,
as the authors of the article, we are very grateful for the reliable review and valuable comments on the content of our paper. We would also like to thank you for the valuable time you devoted to reviewing our article.
We have added a new subsection to the article, '5. Discussion', in which we attempt to provide a cause-and-effect sequence showing how impaired iron metabolism can lead to chronic fatigue.
Thank you very much for every important suggestions. All comments helped us to improve the quality of our article. We have tried to follow all the comments of Reviewers, trying to increase the value of our paper. We hope that we have met your expectations regarding the revision of our article.
Sincerely yours,
Authors.
Round 3
Reviewer 2 Report
By adding an explanatory paragraph in the discussion of the potential molecular origins of fatique in iron metabolism disorders, the authors greatly enhanced the interest and understanding of the subject treated in the article.